# The Influence of Mattress Stiffness on Spinal Curvature and Intervertebral Disc Stress—An Experimental and Computational Study

**DOI:** 10.3390/biology11071030

**Published:** 2022-07-08

**Authors:** Tommy Tung-Ho Hong, Yan Wang, Duo Wai-Chi Wong, Guoxin Zhang, Qitao Tan, Tony Lin-Wei Chen, Ming Zhang

**Affiliations:** 1Department of Biomedical Engineering, Faculty of Engineering, The Hong Kong Polytechnic University, Hong Kong 999077, China; tommyth.hong@connect.polyu.hk (T.T.-H.H.); yawang@polyu.edu.hk (Y.W.); duo.wong@polyu.edu.hk (D.W.-C.W.); guo-xin.zhang@connect.polyu.hk (G.Z.); matthew.tan@connect.polyu.hk (Q.T.); tony.l.chen@connect.polyu.hk (T.L.-W.C.); 2The Hong Kong Polytechnic University Shenzhen Research Institute, Shenzhen 518057, China

**Keywords:** spinal curvature, supine position, curvature measurement, mattress stiffness, different hardness mattresses, finite element method, musculoskeletal, sleep health, intervertebral disc loading

## Abstract

**Simple Summary:**

Due to technical limitations, few studies on the supine position have focused on spinal curvature measurement. This study used a novel technique to measure spinal curvature in the supine position and obtain intervertebral disc pressures through computational simulation. The results revealed an increased craniocervical height when lying on a soft mattress, leading to significantly increased cervical disc loading. The lumbar lordosis was reduced when using the hard mattress, and contact pressure significantly increased. Thus, a medium mattress appeared to be more suitable to use. In cases where a soft mattress is used, the use of a softer or thinner pillow should be considered.

**Abstract:**

Sleeping support systems can influence spinal curvature, and the misalignment of the spinal curvature can lead to musculoskeletal problems. Previous sleep studies on craniocervical support focused on pillow variants, but the mattress supporting the pillow has rarely been considered. This study used a cervical pillow and three mattresses of different stiffnesses, namely soft, medium, and hard, with an indentation load deflection of 20, 42, and 120 lbs, respectively. A novel electronic curvature measurement device was adopted to measure the spinal curvature, whereby the intervertebral disc loading was computed using the finite element method. Compared with the medium mattress, the head distance increased by 30.5 ± 15.9 mm, the cervical lordosis distance increased by 26.7 ± 14.9 mm, and intervertebral disc peak loading increased by 49% in the soft mattress environment. Considering that the pillow support may increase when using a soft mattress, a softer or thinner pillow is recommended. The head distance and cervical lordosis distance in the hard mattress environment were close to the medium mattress, but the lumbar lordosis distance reduced by 10.6 ± 6.8 mm. However, no significant increase in intervertebral disc loading was observed, but contact pressure increased significantly, which could cause discomfort and health problems.

## 1. Introduction

About eight hours of quality sleep daily is essential for maintaining health. A reasonable sleeping support system composed of a mattress and pillows is critical for good sleep. The human spine should be kept in its natural shape during sleep to minimize the internal loading of soft tissues and allow the intervertebral discs (IVD) to rehydrate [1,2,3]. Spinal alignment plays a critical role in sleep health, as spine malalignment may cause neck pain [4,5] or low-back pain [6,7]. Several studies investigated the observation of spinal curvature in the lateral position using geometrical instruments [3,8]. Supine sleep is suggested for “good posture” [9], but the covered back geometry increases the difficulty of conducting spinal curvature measurements [1]. In the supine position, there exist limited tools for spinal curvature measurements. X-rays, CTs, and MRIs can capture spinal curvature in unexposed conditions of the back and are commonly used in medical procedures for spinal disorders [10,11]. However, they have disadvantages and limitations, including radiation exposure to participants, high cost, and the inability to put the sleeping support system into the medical imaging system.

The stiffness of the mattress can affect the alignment of the spine. If the mattress is too firm, the lower back cannot reach the mattress when lying down, leading to an abnormal spine kyphosis. If the mattress is too soft, it can cause the hip area where the weight is concentrated to sink which can lead to an unnatural kyphosis of the lumbar spine, with an over-compression of the IVD front side [12]. However, the described changes in spine shape have rarely been depicted in detail or numerically measured. Most mattress studies focus on the thoracic, buttocks, and limbs but excluded the craniocervical region [1,2,13]. For example, Leilnahari et al. [14] studied the effect of different firmness mattresses on spinal alignment in lateral sleeping positions, but the study only involved vertebrae C7-L5 and excluded the craniocervical region. An investigation of the craniocervical support provided by mattresses is necessary.

Good support for the head and neck is considered an integral part of overall sleep quality [15]. Researchers agree that maintaining the natural lordosis curve of the cervical spine is essential for increasing the time spent in deep sleep [16]. Poor cervical posture during sleep can place pressure on the cervical spine structures. In addition to affecting sleep quality, it may also cause pain, including neck and shoulder pain, tension headaches, and muscle stiffness. [17,18]. However, mattress studies generally focus on low back pain [1,2,13] and rarely concern the head and neck regions. One possible reason is that the head and neck are supported directly by the pillow, which is thought to be less influenced by the mattress.

Previous studies developed several finite element (FE) models of the human body for biomechanical analyses. These include neck ligament models [19,20,21,22], and lumbar spine models [23,24]. These models are mainly used in quasi-static load scenarios to study the internal changes of the spine when subjected to external loads. In recent years, FE models with an integrated human body and sleeping support system have been applied to sleep analysis. Yoshida [25] developed a 2D FE model to analyze the impact of mattresses on sleep comfort. Lee et al. [26] developed FE models based on the human contour obtained by the 3D scanner and used them to analyze the contact pressure in the supine position. The FE models used in these models were highly simplified. The human body comprises complex structures, e.g., the musculoskeletal system includes bones, muscles, tendons, ligaments, and soft tissues. The vertebral column provides rigidity and mobility to the trunk, while the rib cage and pelvis create greater trunk stiffness [12]. FE models with more detailed anatomical structures that take into account the geometry of IVDs may provide additional evidence for the objective representation of mattress comfort [1]. However, computational simulations usually employ oversimplified models to estimate spinal curvature.

This study aimed to investigate the biomechanical effects of the human body, which is interactive with different stiffnesses of mattresses.

The contributions of this study are summarized as follows:Measurement of spinal curvature while lying in a supine position on different-stiffness mattresses.Computational simulation of contact pressure distribution.Investigating the internal stress of IVDs.Investigating the relationship between spinal curvature and musculoskeletal health.

## 2. Materials and Methods

### 2.1. Physical Experiment

#### 2.1.1. Participants

Sixteen participants (29.4 ± 5.1 years) with no history of spinal injuries, deformities, or pathologies were recruited at the university campus. All participants could remain standing for more than 5 min with their arms extended horizontally. The patients’ average height, weight, and BMI were 170.3 ± 7.8 cm, 64.8 ± 13.6 kg, and 22.2 ± 3.4. This study was reviewed and approved by the University. The reference number for ethical approval is HSEARS 20201223001. All participants were informed of the experimental procedures and provided their written informed consent.

#### 2.1.2. Materials

Three polymeric foam mattresses were provided by a mattress manufacturer (C122A/C123B/C105H, Sinomax, Shenzhen, China). The manufacturer considered them representatives of a soft mattress (SM), a medium mattress (MM), and a hard mattress (HM). The materials’ 25% compression indentation load deflection was SM: 20 lbs, MM: 42 lbs, and HM: 120 lbs. The three mattresses had the same dimensions, with a size of 1900 × 1200 × 200 mm. A typical cervical pillow (TDI553511, Infinitus, Guangzhou, China) was adopted in this study to establish a realistic sleep environment. The design and dimensions of the mattresses and the pillow are shown in Figure 1a.

To minimize shipping and storage space, mattress manufacturers typically perform the foaming process of polymer foams by themselves. Thus, most of the mattress foams are non-standard. A plastic tensile test machine (Instron 5569, Instron, MA, USA) measured the stress–strain properties of foams to support the computational simulation. Poisson’s ratio was obtained by measuring the material diameter during the stress–strain test. The testing results are shown in Figure 1b; the strain-stress curves show the hyperelastic properties of typical foam materials [27].

#### 2.1.3. Equipment

A self-developed curvature measurement tape (CMT) [28] was used to measure spinal curvature. This device was validated to measure spinal curvature in the supine position. The length of the sensor tape was 870 mm, which was long enough to cover the entire spine. The sensor tape integrated 30 accelerometers and was able to measure 29 arcs or lines and combine the arcs or lines into a curvature. This sensor tape was thin and flexible with limited interference with sleeping posture. The dimensional accuracy was approximately 99% when measuring a ø200 mm circle [28].

#### 2.1.4. Experimental Procedures

As shown in Figure 2a, the cervical pillow rested on one of the mattresses; the CMT was placed on the centerline of the mattress and pillow for spinal curvature measurement. Figure 2b shows the graphical user interface with the measured curvature. A video demonstration of the data collection process using the curvature measurement tape can be found in Appendix A. Considering that the lack of deformation of the hard mattress could result in a gap between the lumbar lordosis and the mattress top, a soft foam supporter was placed under the CMT to ensure the CMT came into contact with the participant’s back. Participants were topless or wearing tights, lying on the sleeping support system in a supine position. The limbs were placed parallel to the body in the sagittal plane. After the participant’s posture was stabilized, the CMT measured the spinal curvature, and numerical data were saved on a computer. Data collection was performed on three mattresses in a randomized crossover trial. The output data contained 29 arcs or straight lines, each element containing three points (X_start_, Y_start_, X_mid_, Y_mid_, X_end_, and Y_end_) representing the spinal curvature. In the post-processing stage, the data were loaded into CAD software (AutoCAD R13, Autodesk, CA, USA) for analysis.

Figure 3 shows the definition of the spine parameters. A straight line connected the tangent points of thoracic lordosis and sacral lordosis; this line represents the torso inclination (TI) in the sagittal plane. The distance between the head to the TI was defined as the head distance (HD); the distance of cervical lordosis to the TI was measured and defined as the cervical lordosis distance (CLD); and the distance of lumbar lordosis to the TI was defined as lumbar lordosis distance (LLD).

### 2.2. Computational Simulation

An FE model of a male body with details of vertebrae, IVDs, major ligaments of the spine, and bulk tissue of the whole body was developed. The modeling procedures included biomedical imaging, geometric reconstruction, material properties, model assembly, loading, boundary conditions, and model validation.

#### 2.2.1. Participant Information

An adult male without spinal disease was recruited for the model reconstruction. He was 37 years old, 176 cm in height, and 74 kg in weight. Figure 4 illustrates the process from medical image acquisition to mesh generation for the FE simulation.

#### 2.2.2. Geometry Reconstruction

The geometry of the body contour, skull, vertebrae from C1 to L5, sacrum, pelvis, ribs, shoulder blades, and primary legs structures was acquired with an MRI scanner (Philips Achieva 3.0T, Philips, Andover, MA, USA). Before the scan, a custom-made orthosis was fabricated to replicate the contours of the participant’s rear head and back of the torso in an upright position. The orthosis and limb supporters were placed on the scan bed of the MRI scanner. The participant laid on the orthosis and supporter to maintain his body profile and spinal alignment, which were scanned similarly to an an upright standing posture.

The images were segmented and processed in Mimics (Mimics V19, Materialize, Leuven, Belgium) and 3-matics (3-matics V7.0, Materialize, Leuven, Belgium) software. Bones, IVDs, and bulk soft tissue geometries were reconstructed as three-dimensional solid parts. Their locations and configurations were established based on the medical images and anatomy atlas [29,30,31]. The completed model included 34 bones, 24 IVDs, rib cage, brain, and whole-body bulk soft tissue.

#### 2.2.3. Ligaments

This FE model contained the major spinal ligaments, including the anterior longitudinal ligament (ALL), posterior longitudinal ligament (PLL), Flava ligament (FL), cervical ligament (NL), interspinous ligament (ISL), transverse ligament (IL), and capsular ligament (CL). All ligaments were created in the form of the truss. The mechanical properties of the ligaments were obtained from the literature and are listed in Table 1.

#### 2.2.4. Sleeping Support System

Three sleeping support systems were constructed according to the experimental setup. The material properties of mattresses and the pillow were obtained from tensile tests. The hyperelastic mechanical properties under compression were entered into the FE material database. A non-deformable discrete rigid two-dimensional plane was created as the ground plate to support the mattress, pillow, and human model. The dimensions of the ground plate was 2000 × 2000 mm.

#### 2.2.5. Mesh

The mesh creation process was performed with Abaqus 6.14 (Simulia, Dassault Systemes, France). Solid parts including the skull, brain, 24 vertebrae, 24 IVDs, sacrum, pelvis, blades, femurs, shanks, and feet were meshed using linear tetrahedral elements (C3D4). Four hundred and eighty-six trusses were modeled as ligaments to connect bony structures and the bulk tissue of the whole body; all ligaments were meshed using two-node truss elements (T3D2). Hexahedral elements (C3D8) were used to mesh the pillow, mattress, and ground plate. If the mesh is too large, it may decrease the computational accuracy, while a mesh that is too small may lead to a significant consumption of computer power and result in a long computational time. Since cervical discs have a smaller volume than other discs [37], the lower cervical region is the location of most degenerative cervical myelopathy reports [38]. The mesh convergence test was performed in an axial loading condition of the C4-C5 vertebrae, and the IVD between the vertebrae. The mesh size was refined in a repeated simulation at 10% intervals from 5 mm to 2.65 mm. The mesh size of the vertebrae and IVDs was determined to be 3 mm because the deviation of the prediction results was less than 5% compared with the previous results [39].

#### 2.2.6. Material Property

The material properties used in the FE model are listed in Table 1. The assigned materials for the human model were assumed to be homogeneous in all parts. Hyperelastic materials were assigned to the sleeping support system, and the material properties were obtained as presented in Section 2.1.2.

#### 2.2.7. Boundary and Loading Conditions

In this study, the ground plate was locked, and the mattress above the ground plate had all degrees of freedom and could be deformed in any direction. The bottom side of the cervical pillow was in contact with the mattress top at a friction coefficient 0.6. The human model was placed above the sleeping support system, and uniform gravity (g = 9.81) was applied downward to the ground plate during the simulation (Figure 5).

#### 2.2.8. Model Validation

This FE model was validated by comparing spine parameters of the experimental measurements of lying in a supine position on different-stiffness mattresses and the computational results. As shown in Figure 6, the correlation analysis indicated a significant linear relationship between the experiment value and the computational answer (r = 0.88; *p* < 0.001). A Pearson’s correlation coefficient of between 0.7 and 1.0 indicated a strong positive linear relationship through a firm, linear rule [40].

## 3. Results

### 3.1. Spinal Curvature

This study used CMT to measure the spinal curvatures of participants lying in a supine position in SM, MM, and HM environments. The CAD software extracted the spine parameters, including TI, HD, CLD, and LLD. The experimental results are shown in Figure 7. There were significant differences in the TI. Among the three mattresses, the average TI in the HM environment was of the smallest value (TI = 1.56 ± 0.63°), indicating that the body was almost parallel to the mattress top. A moderate TI value was measured in the MM environment (TI = 3.64 ± 1.09°). The highest TI value was measured in the SM environment (6.86 ± 4.85°), where the buttock was sunken more from the top of the mattress. It was observed that HD and CLD had a similar pattern, where the SM environment showed the largest HD and CLD averages (SM: HD = 85.8 ± 21.5 mm, CLD = 96.6 ± 18.3 mm), and the MM and HM results were both smaller and similar (MM: HD = 55.3 ± 8.9 mm, CLD = 69.9 ± 6.8 mm; HM: HD = 60.0 ± 11.0 mm, CLD = 70.2 ± 6.3 mm). For LLD, the results of both SM and MM were relatively similar, while the results of HM were significantly smaller (SM: LLD = 22.9 ± 6.5 mm; MM: LLD = 22.2 ± 4.8 mm; HM: LLD = 11.6 ± 5.2 mm).

### 3.2. FE Predicted Pressure Distribution

Contact pressure measurement systems aim to use thin and flexible sheet sensors to minimize mattress support interference [41,42]. However, the sensor may disperse the concentrated pressure and thus underestimate the peak pressure [1]. Thus, the FE method has gained popularity as a method by which to obtain contact pressure and is widely used [26,43]. The computational result of contact pressure between the human model and the sleeping support system is shown in Figure 8. The maximum regional contact pressure for several body regions, including the occiput, cervical, scapula, buttocks, calves, and heels, is marked on the figure. In general, the SM environment exhibited the lowest peak contact pressure (occiput = 7.5 kPa, cervical = 9.7 kPa, scapula = 4.5 kPa, buttock = 5.9 kPa, calve = 3.3 kPa, heel = 6.9 kPa). Most regions of the MM environment (occiput = 7.7 kPa, cervical = 7.7 kPa, scapula = 7.2 kPa, buttock = 8.9 kPa, calve = 5.3 kPa, heel = 10.1 kPa) were higher than the SM, and the predicted peak contact pressure for the HM environment was significantly higher than the SM and MM environments (occiput = 8.9 kPa, cervical = 5.2 kPa, scapula = 24.3 kPa, buttock = 28.6 kPa, calve = 7.4 kPa, heel = 17.9 kPa).

The contact area was obtained using computational simulations (SM: pillow = 272 cm^2^, mattress = 3962 cm^2^; MM: pillow = 254 cm^2^, mattress = 2974 mm^2^; HM: pillow = 251 cm^2^, mattress = 1915 cm^2^). The metrology criterion was based on the area where the pressure was higher than 1 kPa.

### 3.3. IVD Peak Loading

The computational simulation provided the peak loading of the IVDs in the supine position. The predicted values and distribution of Mises stress are graphically presented in Figure 9a, and the peak pressure of individual IVD is shown in Figure 9b.

## 4. Discussion

### 4.1. Findings

When lying on mattresses of different stiffness in a supine position, the participants’ heads was kept at a similar height, while the torso sank into the mattress with significant differences. Figure 10a shows that the body sunk into the mattress the most in the SM environment. A greater TI value indicates that the pelvis sinks into the mattress more than the chest area. In contrast, the body sunk the least in the HM environment, with the smallest TI value measured, in which the torso tended to be horizontal. The sinking of the body brings about changes in the spinal curvature. A set of spinal curvatures measured with CMT are shown in Figure 10b for reference.

#### 4.1.1. Changes in the Craniocervical Region

Compared with the MM environment, the experimental results showed that the head was raised by 30.5 ± 15.9 mm in the SM environment. The CLD also increased by 26.7 ± 14.9 mm. The increase in craniocervical height resulted in a significant rise in IVD loading. In the SM environment, the peak loading of IVD C5-C6 was 316 kPa, which was 49% higher than the loading of the same IVD in the MM environment (MM: IVD_C5-C6 = 212 kPa). Previous studies indicated that a high IVD loading may interfere with IVD rehydration and cause neck pain [4,5]. Therefore, increasing the head and cervical height may lead to musculoskeletal health risks.

Although the same pillow was used for the experiments, the craniocervical region appeared to have stronger support in the SM environment. Craniocervical loading was applied to the pillow top; the load then spread inside the foam pillow and formed a larger area at the pillow bottom to support the same loading (Figure 11). The deformation of the craniocervical region projected onto the mattress was relatively reduced due to the loading dispersion. From the material test results (Figure 1b), the mechanical property of the cervical pillow was found to be similar to the MM, while the SM was much softer. In this case, the stiffer pillow acted as a hard plate to spread the load on the head and neck. The contact area of the craniocervical region was 272 cm^2^, and the area of the pillow bottom was 55 x 35 = 1925 cm^2^. It can be considered that the area supporting the craniocervical site increased significantly. As a result, the area of the mattress under the pillow experiences less deformation due to the reduced pressure.

The torso sinks deeper into the softer mattress, but the craniocervical area sinks less, which results in high IVD loading. As can be seen in Figure 10c, we believe that a softer or lower pillow can reduce the craniocervical height and lessen IVD loading. However, further studies should be conducted to obtain the appropriate pillow height for the user.

#### 4.1.2. Changes in Lumbar Lordosis

The LLD measured in the HM environment was 10.6 mm lower than that in the MM environment (*p* < 0.001), while the LLD values for the MM and SM environments were close, with no significant differences (*p* > 0.05). Variations in LLD may result in different IVD loading distributions. The peak loading of the IVD was calculated using the FE method, as shown in Figure 9. By comparing the lumbar IVDs from T12 to the sacrum, the peak loading was found to be 119 kPa in the HD environment, 117 kPa in the MM environment, and 93 kPa in the SM environment. Although slightly higher, the HM environment was still within the normal lying range of nucleus pressure [44], and the peak loading of lumbar IVDs was much lower than that of cervical IVD. Therefore, the change in LLD is considered acceptable.

#### 4.1.3. Soft Tissue Deformation

The FE results show that the contact pressures in the HM environment were mostly higher, especially in the upper back and buttock regions. Peak pressure typically increased by 3–4 times compared to the MM environment. The high contact pressure can cause discomfort and skin sores [45]. Therefore, the HM was considered suboptimal. However, head and neck support appeared to be beneficial because the IVD peak loading remained low.

The SM environment exhibited the largest contact area in the FE prediction, where the peak pressures were generally lower. The only exception was the cervical site, where the peak pressure in SM was higher than the HM and MM environment values. In response to the findings of Section 4.1.1, the elevation of the craniocervical region produced higher contact pressures in the cervical region. Previous studies suggested that sleep quality is improved when sleeping on a system with a moderate pressure distribution [41]. Therefore, the MM environment can be considered as appropriate.

#### 4.1.4. Insufficient Upper Back Support

The sleeping support systems did not adequately support the upper back, for which the overhang area is shown in Figure 12. The height of the cervical pillow was 90 mm (Figure 1a), and this 90mm cervical raise left the upper back empty. This finding could lead to product development to improve upper back support.

### 4.2. Limitations

The sample size was calculated using a sample size calculator (G*power 3.1.9.4, Heinrich-Heine-University, Düsseldorf, Germany), and the test effect size was moderate power with a significance level of 5%. However, the participants were recruited on college campuses and did not reflect the population. A larger sample size considering different age groups might allow the investigation of ageing discs, ligaments, and tendons.

The human body contains complex biomechanical structures, including bones, muscles, tendons, ligaments, fascia, and skin. To enhance the accuracy of the computational simulation, this FE model involved the skull, vertebrae, IVD, ligaments, and surrounding soft tissue. However, it did not consider the muscles and assumed they were relaxed in the supine position. It is undeniable that some involuntary muscles stabilize the spine in the supine position.

Quadratic tetrahedral elements could improve the mesh convergence of the simulation. However, we did not apply the quadratic tetrahedral elements because of the high computing power required. Previous studies found that models meshed with quadratic tetrahedral elements required costly computational power, increasing the computation time by about three times [46]. Since this FE model contained many complex parts, linear tetrahedral elements were adopted.

### 4.3. Future Works

We focused on comparing mattresses of different stiffnesses in this study. There are various types of mattresses, such as spring, foam, and airbag mattresses. The shape of the pillow is a critical element of pillow design as it contributes to the level of neck support and the user’s overall comfort. Additionally, the sleeping support can be produced from various materials, such as cotton, polyester, foam, feather, and latex. Different materials impact both sleep time and sleep quality [15]. Future studies may consider evaluating the differences between different types or brands of sleeping supports.

Though males and females have different body physiques, a previous study demonstrated that there was no significant interaction between gender and pillow height on craniocervical pressure [17]. Additionally, there are scarce gender-specific sleeping support systems on the market, so this study did not take gender into account. Therefore, this study did not compare the effects of sleep support systems on gender. On the other hand, our study focused on adults and the findings do not apply to children. The effects of sleep support systems on children are worthy of study because children are in a period of rapid growth, and sleeping support systems may significantly impact such growth.

## 5. Conclusions

This study combined experimental measurements and computational simulation analyses to evaluate the influence of mattress stiffness on the musculoskeletal system. Using the same pillow, while changing the mattress stiffness from medium to soft, the cervical lordosis increased by 26.7 ± 14.9 mm, head distance increased by 30.5 ± 15.9 mm, and there was no significant change in lumbar lordosis. The increase in cervical lordosis resulted in a 49% increase in IVD peak loading. When comparing the medium and hard mattresses, cervical lordosis did not change to a considerable degree, but lumbar lordosis decreased by 10.6 ± 6.8 mm. The decrease in lumbar lordosis caused a slight increase in IVD peak loading, but this is not deemed a critical problem. However, the significant increase in contact pressure with a hard mattress can cause discomfort and health problems. When using a soft mattress, it should be noted that the pillow’s support may be magnified, and it is recommended to choose a softer or thinner pillow.

## Figures and Tables

**Figure 1 biology-11-01030-f001:**
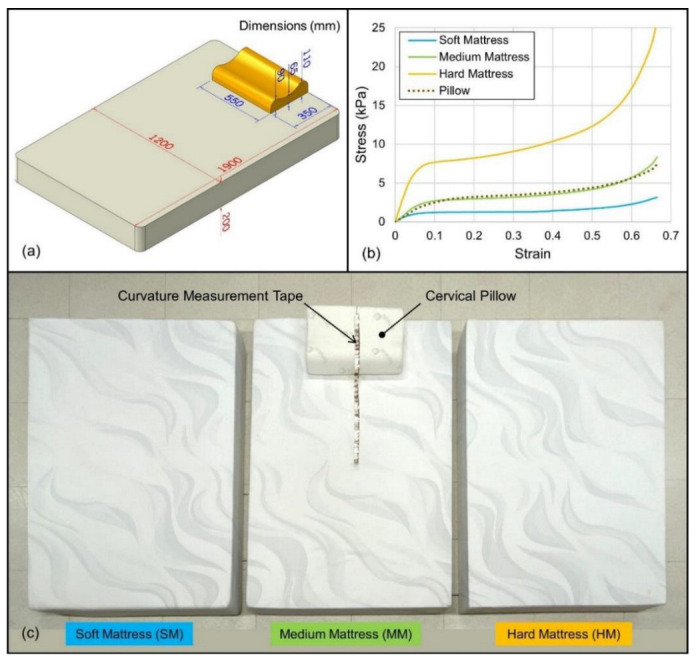
Sleeping support systems and material properties. (**a**) Dimensions of the mattress and cervical pillow. (**b**) Compressive stress–strain curves for mattress and pillow materials. (**c**) The placement of the three mattresses used for this experiment, with the cervical pillow and curvature measurement tape placed on the medium mattress.

**Figure 2 biology-11-01030-f002:**
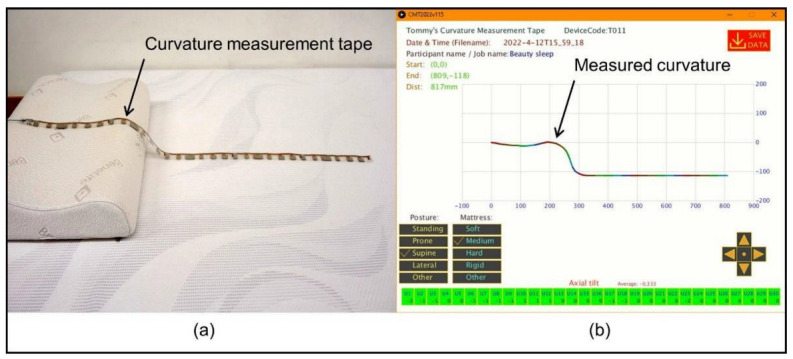
The setup of curvature measurement tape (**a**) The sensor tape placed on the sleeping support system; (**b**) The graphical user interface.

**Figure 3 biology-11-01030-f003:**
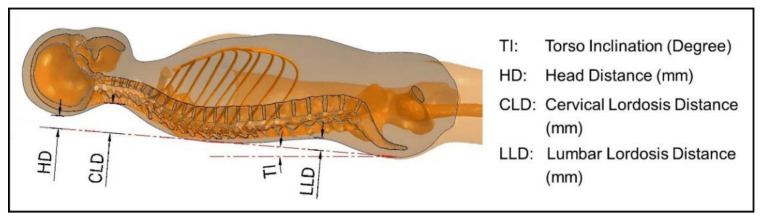
The definition of spine parameters.

**Figure 4 biology-11-01030-f004:**
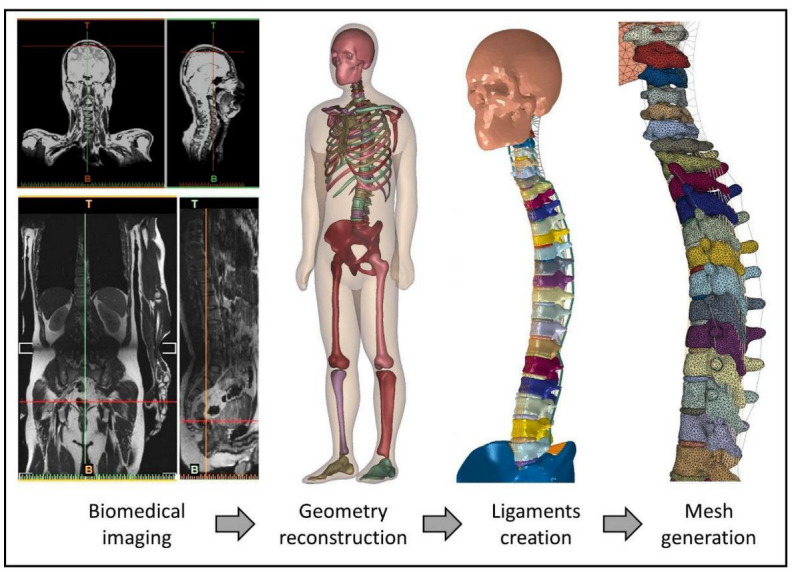
Flow diagram illustrating the model creation process includes biomedical image acquisition, geometry reconstruction, ligaments creation, and mesh generation for finite element simulation.

**Figure 5 biology-11-01030-f005:**
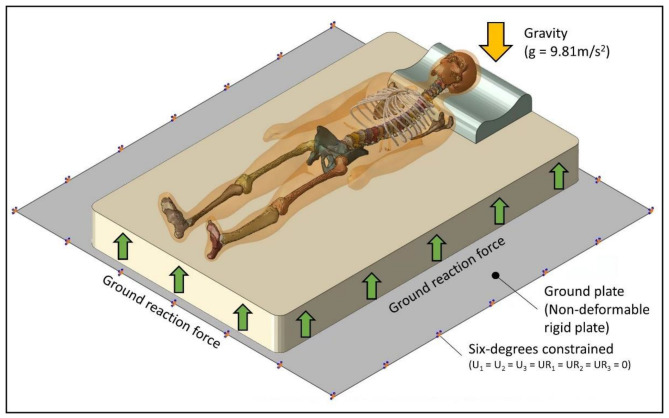
Model geometry and the configuration of boundary and loading conditions.

**Figure 6 biology-11-01030-f006:**
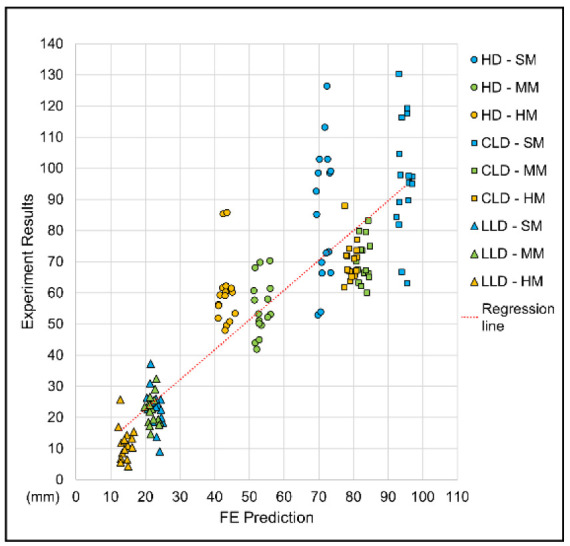
Jitter plot for correlation analysis between finite element predictions and experimental results for model validation.

**Figure 7 biology-11-01030-f007:**
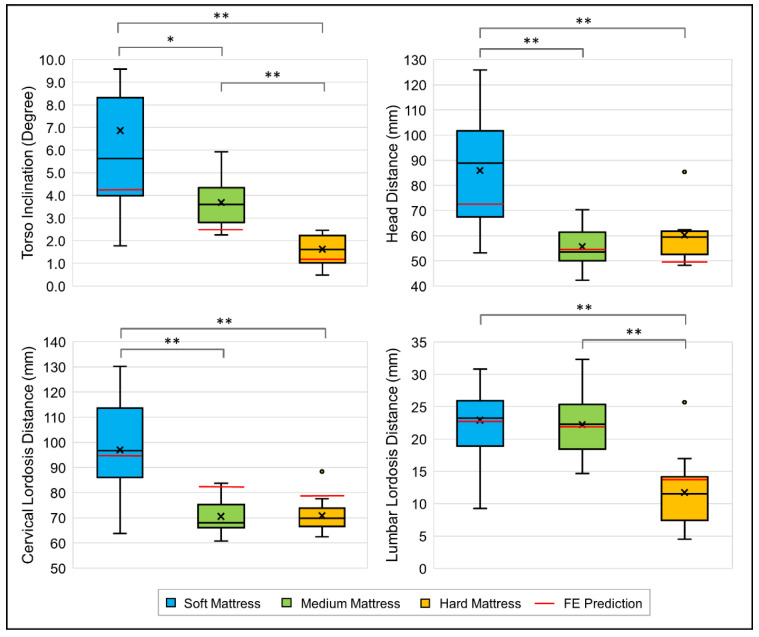
Box plot of experimental results and finite element predictions of spine parameters; * indicates *p* ≤ 0.05, ** indicates *p* ≤ 0.001, and ° indicates an outlier.

**Figure 8 biology-11-01030-f008:**
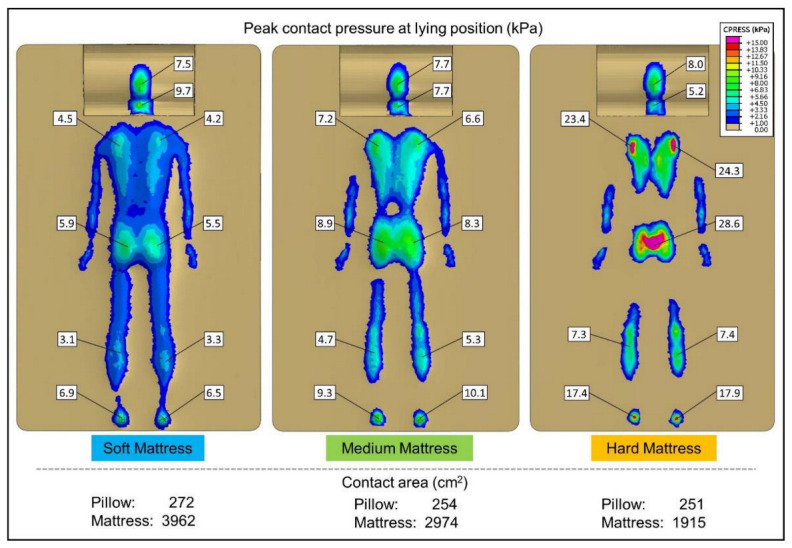
Predicted pressure distribution and contact area by finite element method.

**Figure 9 biology-11-01030-f009:**
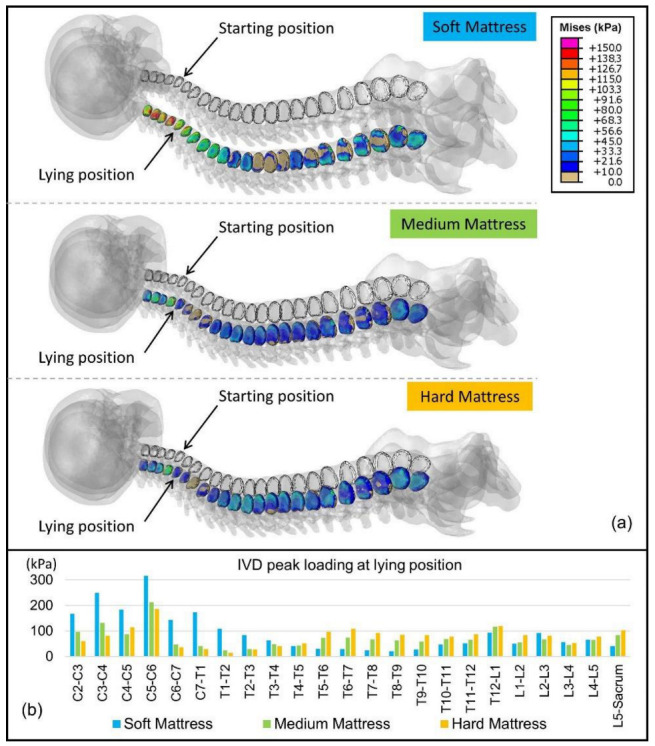
The graphical plot of intervertebral disc loading (**a**) The distribution of intervertebral disc loading. (**b**) The peak loading of the intervertebral disc at a lying position.

**Figure 10 biology-11-01030-f010:**
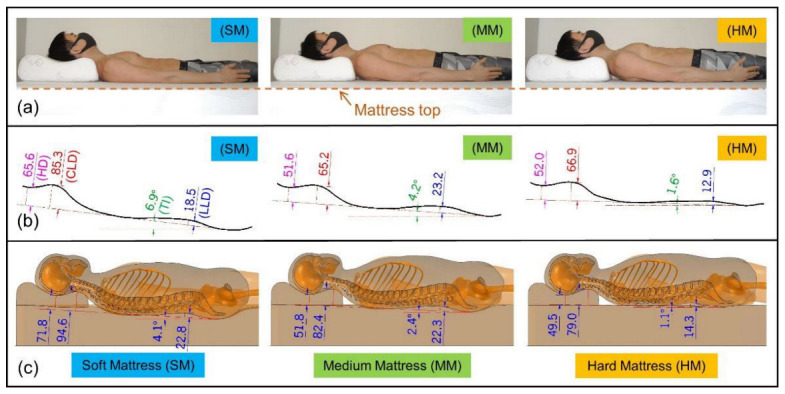
The experimental and computational results (**a**) A side view of lying on the sleeping support system with different stiffness mattresses; (**b**) The measured spinal curvatures by the CMT; (**c**) The sectional side view of the finite element model.

**Figure 11 biology-11-01030-f011:**
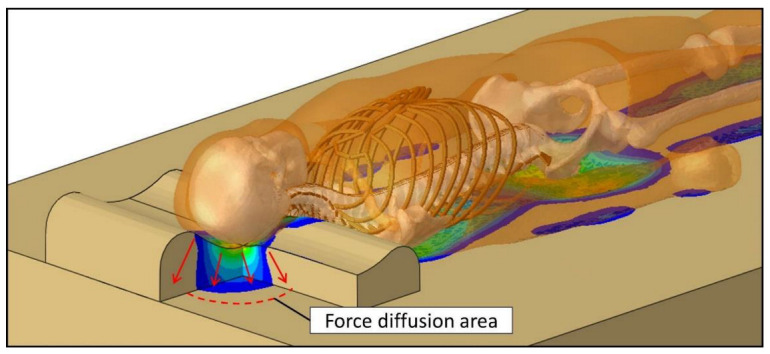
Illustration of force diffusion inside the pillow.

**Figure 12 biology-11-01030-f012:**
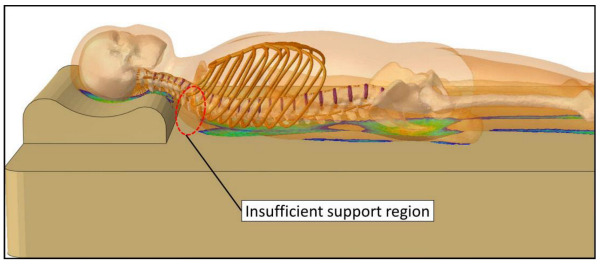
Illustration of insufficient support region.

**Table 1 biology-11-01030-t001:** Material properties used in the finite element model.

Part	Material Property	References
Bone	E = 12 GPa, v = 0.3	[32]
Rib cage	E = 4.86 GPa, v = 0.3	[33]
Encapsulated soft tissue	E = 15 kPa, v = 0.49	[34]
*Ligaments*		[21,35,36]
Anterior longitudinal ligament	E = 11.9 kPa, v = 0.39	
Capsular ligament	E = 7.7 kPa, v = 0.39	
Flava ligament	E = 2.4 kPa, v = 0.39	
Interspinous ligament	E = 3.4 kPa, v = 0.39	
Intertransverse ligament	E = 10 kPa, v = 0.39	
Posterior longitudinal ligament	E = 12.5 kPa, v = 0.39	
*Foam materials*		
Soft mattress		
Medium mattress	Figure 1b, v = 0.01	
Hard mattress		
Pillow		
Ground plate	Rigid part	

## Data Availability

The data presented in this study are available on request from the corresponding author. The data are not publicly available due to privacy issues of the participants.

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
