# Peer review of "The Influence of Mattress Stiffness on Spinal Curvature and Intervertebral Disc Stress—An Experimental and Computational Study"

_biology, 2022, doi:10.3390/biology11071030_

Round 1

Reviewer 1 Report

I'm very glad to become acquainted with such interesting research. Hope that my review will help to improve the quality of the article. The authors made significant work, but, still, there are some notes:

The abbreviations should be deleted from abstract.

There are some misprints like using comma to separate decimal part. In general the research is well written and well designed.

I advise authors to use quadratic tetrahedral elements - it will improve the mesh convergence.

It was very exciting to become acquainted with your research.

Author Response

I'm very glad to become acquainted with such interesting research. Hope that my review will help to improve the quality of the article. The authors made significant work, but, still, there are some notes:

Q1) The abbreviations should be deleted from abstract.

Response: Thank you for your appreciation. We remove the abbreviation and retain the long-form accordingly.

Q2) There are some misprints like using comma to separate decimal part. In general the research is well written and well designed.

Response: Thank you. The commas have been corrected accordingly.

Q3) I advise authors to use quadratic tetrahedral elements - it will improve the mesh convergence.

Response: Thank you for the useful comment. The quadratic tetrahedral elements could improve the mesh convergence but impose a high demand on computing power. We address this issue in the limitation.

Revision: Lines 380-385 (in "simple markup" mode), it is also attached here:

Quadratic tetrahedral elements could improve the mesh convergence of the simulation. However, we did not apply the quadratic tetrahedral elements because of the high demand for computing power. Previous studies have found that models meshed with quadratic tetrahedral elements required expensive computational power, increasing the computation time by about three times [47]. Since this FE model contains many complex parts, linear tetrahedral elements were adopted.

Reviewer 2 Report

Overall, this manuscript is well written down. This is a good body of work and worthy of publication with only a minimal amount of editing is not a doubt in my mind. But I am wondering if authors did experiments with mattress from different manufactures and with different type of pillows. Did authors consider testing the effects of mattress and pillow to different genders, since male and female have different body features? What influences will have to kids who are under growing fast and what type of mattress would be suggested for kids? How to define a good sleep time and sleep quality in your experiment? Authors suggest using a thinner and softer pillow if using a soft mattress, but how thin or how soft? I would suggest authors to give some definitive numbers for thickness or softness, similar problems also happen to other parameters.

Author Response

Reviewer 2

Overall, this manuscript is well written down. This is a good body of work and worthy of publication with only a minimal amount of editing is not a doubt in my mind.

Q1) But I am wondering if authors did experiments with mattress from different manufactures and with different type of pillows.

Response: Thank you for the comments. We have selected beddings from one particular manufacturer. We agree that there might be some variations between manufacturers and we are highlighting this issue in the discussion.

Revision: Lines 384-391 (in "simple markup" mode), it is also attached here:

We focused on comparing mattresses of different firmness in this study. There are various types of mattresses, such as spring, foam, and airbag mattresses. The shape of the pillow is a critical element of pillow design as it contributes to the amount of neck support and the user's overall comfort. Additionally, the sleeping support can be produced from various materials, such as cotton, polyester, foam, feather, and latex. Different materials impact both sleep time and sleep quality [15]. Future studies may consider evaluating the differences between different types or brands of sleeping supports.

Q2) Did authors consider testing the effects of mattress and pillow to different genders, since male and female have different body features?

Response: This is an interesting point. We assumed that there was no gender difference on the effects based on an existing study. We supplement the information in the paper.

Revision: Lines 392-396 (in "simple markup" mode), it is also attached here:

Though males and females have different body physiques, a previous study has shown that there was no significant interaction between gender and pillow height on craniocervical pressure [17]. Also, there are fewer gender-specific sleeping support sys-tems on the market, so this study did not take gender into account. Therefore, this study did not compare the effects of sleep support systems on gender.

Q3) What influences will have to kids who are under growing fast and what type of mattress would be suggested for kids?

Response: This is an exciting point. We did not target the children population in this study yet. It is good to be in our future study.

Revision: Lines 396-399 (in "simple markup" mode), it is also attached here:

On the other hand, our study focused on adults and the findings do not apply to children. The effects of sleep support systems on children are worthy of study because they are in a period of rapid growth, and sleeping support systems may significantly impact their growth.

Q4) How to define a good sleep time and sleep quality in your experiment?

Response: This study targeted biomechanical responses and did not consider sleep duration and quality.

Q5) Authors suggest using a thinner and softer pillow if using a soft mattress, but how thin or how soft? I would suggest authors to give some definitive numbers for thickness or softness, similar problems also happen to other parameters. Mattresses and pillows from different manufacturers should be compared too.

Response: We recommend using a thinner or softer pillow if using a soft mattress. However, further experiments and comparisons of more sleeping supports should be conducted to recommend the optimal pillow for the user.

Revision: Lines 329-332 (in "simple markup" mode), it is also attached here:

The torso sinks deeper into the softer mattress, but the craniocervical area sinks less, which results in high IVD loading. From Figure 10(c), we believe that a softer or lower pillow can reduce the craniocervical height and lessen the IVD loading. However, further study should be conducted to obtain the appropriate pillow height for the user.

Reviewer 3 Report

This is an interesting study and the authors have collected a unique dataset using good methodology. The paper is generally well written and structured. The introduction is relevant and theory based. Sufficient information about the previous study findings is presented for readers to follow the present study and procedures. The methods are generally appropriate and provide rationale and practical method of measuring spine status on different mattress. Overall, the results are clear well presented.  Overall, this is a high quality manuscript that has implications for the theoretical ad scientific  basis. Specific comments follow.

433-435 line - please make correction in text

Author Response

This is an interesting study and the authors have collected a unique dataset using good methodology. The paper is generally well written and structured. The introduction is relevant and theory based. Sufficient information about the previous study findings is presented for readers to follow the present study and procedures. The methods are generally appropriate and provide rationale and practical method of measuring spine status on different mattress. Overall, the results are clear well presented.  Overall, this is a high quality manuscript that has implications for the theoretical ad scientific  basis. Specific comments follow.

433-435 line - please make correction in text

Response: Thank you for your appreciation. The reference was checked and modified.

Revision: Lines 463-464 (in "simple markup" mode), it is also attached here:

  1. Horiba, Y.; Kamijo, M.; Inui, S.; Yoshida, H.; Shuimizu, Y., Study on relation between sleeping comfort and sleeping posture. International conference on Kansei engineering and emotion research 2010, PP. 1656-1659.
